

# A robust recurrent ANFIS for modeling multi-step-ahead flood forecast of Three Gorges Reservoir in the Yangtze River

Yanlai Zhou[1,2], Fi-John Chang[1,*], Shenglian Guo[2], Huanhuan Ba[2], Shaokun He[2]

[1] Department of Bioenvironmental Systems Engineering, National Taiwan University, Taipei, 10617, Taiwan, ROC.

[2] State Key Laboratory of Water Resources and Hydropower Engineering Science, Wuhan University, Wuhan, 430072, China.

*Correspondence to*: Fi-John Chang (changfj@ntu.edu.tw).

**Abstract**. Accurate and robust multi-step-ahead flood forecast during flood season is extremely crucial to reservoir flood control. A modified hybrid learning algorithm, which fuses the Least Square Estimator (LSE) with Genetic Algorithm (GA), is proposed for optimizing the parameters of recurrent ANFIS (R-ANFIS) model to overcome the instability and local minima problems as well as improve model's generalization and robustness. A coherent set of evaluation criteria is used to fully explore the model's accuracy (MAE, RMSE, CC & CE) and robustness (reliability, vulnerability & resilience). Three types of ANFIS (i.e. Classic, Recurrent, and Modified Recurrent) models with their optimal input variables identified by the Gamma Test are utilized for modeling multi-step-ahead flood forecast of Three Gorges Reservoir in the Yangtze River, respectively. Taking the horizon t+12 (three days ahead), for example, the comparison analysis between C-ANFIS and R-ANFIS indicates that the R-ANFIS model can largely improve the CE, CC, reliability and resilience by 38.09%, 17.36%, 28.30% & 140.26% as well as significantly reduce the MAE, RMSE, vulnerability by 68.03%, 47.98% & 13.32%. The comparison analysis between R-ANFIS and MR-ANFIS shows that the MR-ANFIS model can further enhance the CE, CC, reliability and resilience by 2.04%, 2.04%, 5.05%, and 3.61%, respectively, as well as decrease the MAE, RMSE, vulnerability by 9.91%, 13.79%, and 9.92%, respectively. Such results evidently promote data-driven model's generalization (accuracy & robustness) and leads to better decisions on real-time reservoir operation during flood season.

**Keywords:** Data-driven model; Modified Recurrent ANFIS (MR-ANFIS); Rainfall-runoff process; Artificial Intelligence (AI); Multi-step-ahead flood forecast



## 1 Introduction

Yangtze River is the longest river in China, where a great many farms and important industrial areas are built beside the river. To control and promote water resource utilization, a series of large dams have been built along the river for multiple purposes, such as hydropower flood control, water supply, and navigation. Among them, the Three Gorges Dam is the largest dam structure in the world and largest hydroelectric project to date, which could generate approximately 90 billion kW·h/year of hydroelectric power for Shanghai and other cities and protect millions of people downstream from flooding. Thus, it is crucial to pay particular attention to reservoir flood forecasting to properly address the flow variability for promoting multi-beneficiary resource management and to gain the highest satisfaction and/or maximal total benefits from all the goals of the reservoir operation. Extremely limited response time to flood disasters in river basin or urban area makes real-time reservoir operation very challenging and reveal an urgent need for accurate and robust multi-step-ahead inflow forecasting models in managing contingencies and emergencies and in alleviating flood risk as well as loss of life and property.

Rainfall-runoff relationship is one of the most popular yet complex practices of data-driven models (Abrahart et al., 2012). Data-driven techniques like artificial neural networks (ANNs) and fuzzy inference systems (FIS) have been widely applied with success to modeling runoff based on rainfall data in operational hydrology (Darras et al., 2015; Chang et al., 2014; Chen et al., 2014; Chen et al., 2013; Yang et al., 2013; De 2013; Mount et al., 2013; Li et al., 2009; Xiong et al., 2004). The FIS is capable of coping with imprecision and uncertainty while ANNs have adaptive learning capabilities of being identified input-output patterns (Wandera et al., 2017; Khan 2016; Sun et al., 2016; Baghdadi et al., 2012; Gunnink et al., 2012; Lohani et al., 2011). Neuro-Fuzzy networks combine the advantages of both fuzzy reasoning and neural networks. The classic Adaptive-Network-based Fuzzy Inference System (C-ANFIS) proposed by Jang (1993) could represent one of the state-of-the-art Neuro-Fuzzy networks and has been widely implemented with satisfaction in streamflow forecasts (Chang and Chang, 2006; Chang et al., 2016; Tsai et al., 2014; Firat and Güngör, 2008; Khac and Hock, 2012).




However, the C-ANFIS as one of feedforward type of Neuro-Fuzzy networks may be insufficient give satisfactory performance for time varying occurrence of rainfall-runoff relationship. Rainfall-runoff simulation is one of the most popular real-world hydrological applications, while they are time dependent dynamic (non-stationary) processes whose input-output patterns cannot be completely identified using static (stationary) ANFIS. Recurrent ANFIS (R-ANFIS) has feedback connection in its topology and its mechanism is dynamic (Zhang and Morris, 1999). Such dynamic mechanism has excellent property that input-output mapping is time-varying rather than a fixed map (long-term memory) and can integrate short-term memory with long-term memory in multi-step-ahead forecast (refs. Juang, 2002; Mastorocostas and Theocharis, 2002), nevertheless this feature and capability of recurrent-ANFIS do not get much attention in the hydrological fields. The original hybrid learning algorithm proposed by Jang (1993) integrate Least Square Estimator (LSE) with steepest or gradient descent algorithms for locally tuning the parameters of C-ANFIS so as to minimize the error between output and target. Whereas the steepest or gradient descent algorithms are derivative optimization algorithms and have the instability, sensitivity to initial conditions, local minima problems (Tamura et al., 2008). Thus, a modified hybrid learning algorithm is proposed here to overcome these defects and improve the generalization and robustness of R-ANFIS.

This study intends to investigate the R-ANFIS for modeling multi-step-ahead flood forecast in comparison to the C-ANFIS, and then modify the original hybrid learning algorithm for enhancing the forecast accuracy and prolonging forecast horizon of R-ANFIS by fusing the LSE with Genetic Algorithm (GA). The GA is used to optimize the premise nonlinear parameters and the LSE is utilized to optimize the consequent parameters of R-ANFIS model. Such modified hybrid learning algorithm provides derivative free exploration for solution in input-output space in comparison to the original hybrid learning algorithm. The paper is organized as follows: Section 2 briefly introduces the study area and materials. Section 3 described the methodology adopted in this study, which comprises three major parts: (1) presenting a theory of the C- ANFIS, R-ANFIS and modified R-ANFIS models adopted in this paper, (2) describing the original and modified hybrid learning algorithms, and (3) elaborating a coherent set of model's performance evaluation metrics. The results and discussion are shown in Section 4, and final remarks are drawn in Section 5.



## 2 Study area and materials

The Yangtze River (Chang-Jiang) has a total length of 6,300 km with a drainage area of 1.80 million km$^2$, and it is the longest river in China (Fig. 1). The cascade reservoirs of Xiangjiaba (XJB) reservoir and Three Gorges reservoir (TGR) are located at the mainstream of Yangtze River from upstream to downstream. The XJB and TGR reservoirs, which were built in 2014 and 2003, respectively, are the pivot hydraulic facilities in the Yangtze River basin for flood control, hydropower generation, navigation, etc. The drainage areas of the XJB and TGR are 0.46 and 1.00 million km$^2$, respectively, and these reservoirs have the flood control capacity of 0.90 and 22.15 billion m$^3$, respectively. The inflow of TGR consists of three components, the main upstream inflow controlled by XJB reservoir, the tributaries inflow controlled by eight flow gauged stations and two (I & II) regional rainfall stations. The information of the reservoirs, rives, streamflow gauged stations and regional rainfall stations in the Yangtze River basin between the XJB and TGR reservoirs can be found in Fig. 1. The observations of XJB reservoir flow, eight flow gauged stations, I & II region rainfalls and TGR flow in the flood season (June 1st to September 30th) from 2003 to 2016 year (14 years) with 6 hours' time-step are available for the TGR inflow forecasting. The observed rainfall data of the I & II region rainfall gauged stations are utilized for calculating the I & II region average rainfall. Take TGR for example, Fig. 2 presents the boxplot of the observed TGR inflow in different years. On the basis of similar statistic values (max, mean, min & derivation) in boxplot (Fig. 2), the data series are divided into the training period (2003 to 2010 year, 9 years) and testing period (2011 to 2016 year, 5 years), which can properly cope with the overfitting problem of data-driven model.

## 3 Methodology

### 3.1 Modeling approach-ANFIS

Developing accurate and robust data-driven model for modeling multi-step-ahead flood forecast is the major goal of this study. Bearing this in mind as a motivation, a modified hybrid-learning algorithm is proposed here by fusing Least Square Estimation (LSE) with Genetic Algorithm (GA) for optimizing the parameters of R-ANFIS. Inflow of TGR is modeled at forecast horizon up to 3 days ahead ($Q_{t+4}$,



$Q_{t+8}$, $Q_{t+12}$) with 6 hours' time-step by using three modeling approaches, and the model architecture is shown in Fig. 3. The Model 1 (C-ANFIS) is selected as benchmark, and its results are compared with those of Model 2 (R-ANFIS) for the purpose of evaluating the performance of recurrent (dynamic) learning mechanism in multi-step-ahead flood forecast. The comparison between Model 2 and Model 3 (Modified R-ANFIS) focuses on how the different hybrid learning algorithms affect the outputs of R-ANFIS models.

For completeness, we firstly present a brief description of the C-ANFIS (Jang, 1993). We assume the ANFIS architecture has two inputs $x_1$ and $x_2$ as well as one output y based on a popular rule set with two if-then rules of Sugeno's type (Sugeno and Kang 1988), and is defined as the following.

$$\text{Rule 1: if } x_1 \text{ is } A_1 \text{ and } x_2 \text{ is } B_1, \text{ then } f_1 = p_1x_1 + q_1x_2 + r_1 \tag{1}$$

$$\text{Rule 2: if } x_1 \text{ is } A_2 \text{ and } x_2 \text{ is } B_2, \text{ then } f_2 = p_2x_1 + q_2x_2 + r_2 \tag{2}$$

where A and B form a fuzzy set. $A_i$ and $B_i$ are the $i$th ($i=1,2$) linguistic term (such as "small" or "large") of the two inputs $x_1$ and $x_2$, respectively. $p_i$, $q_i$ and $r_i$ are the linear parameters in $i$th if-then rules. $f_i$ is the $i$th if-then rules.

The ANFIS consist of five layers (see Fig.2). In the following lines we describe briefly the operation of each layers. For a full description please refer to Jang, (1993).

**Layer 1** (Inputs layer): nodes of this layer perform fuzzification by assigning the membership grade of the linguistic of the input variables. The Gaussian function with two parameters is selected as the default membership function in this study. Parameters in this layer are referred to as premise parameters.

**Layer 2** (Fuzzy-AND operation): each node in this layer performs a fuzzy-AND operation using T-norm operators.

**Layer 3** (Normalization): the output of the $i$h node is the ratio of the $i$th rule' firing strength to the sum of all the fuzzy rules firing strengths:

**Layer 4** (Consequent layer): each node in this layer is an adaptive node with a linear function. Parameters in this layer are referred to as consequent parameters.

**Layer 5** (Output layer): under the Multi-Inputs and Single Output (MISO) pattern, this layer consists of only one node that computes the network's output as the algebraic sum of the node's inputs.





As aforementioned, the set of total parameters in ANFIS models consist of premise (nonlinear) parameters and consequent (linear) parameters. In Fig. 3, the Model 1 (C-ANFIS) is static data-driven model and trained by the original hybrid learning algorithm (LSE & steepest descent algorithm), while the Model 2 (R-ANFIS) is a dynamic data-driven model and also trained by the original hybrid learning algorithm (LSE & steepest descent algorithm). The Model 3 (Modified R-ANFIS) is also dynamic data-driven model and trained by the modified hybrid learning algorithm (LSE & Genetic Algorithm) proposed in this study. The flowchart of original and modified hybrid learning algorithm is described below.

**3.2 Optimizing model parameters by hybrid learning algorithm**

The C-ANFIS and R-ANFIS employ the original hybrid learning algorithm for optimizing parameters, i.e., the steepest descent algorithm is employed in order to optimize the premise parameters of Layer 1, and Least Square Estimator (LSE) is employed so as to optimizing the consequent parameters of Layer 4. To overcome the local minima and instability problems of the original hybrid learning algorithm, a modified hybrid learning algorithm is proposed in this study by fusing the LSE with Genetic Algorithm (GA) for optimizing the parameters of R-ANFIS. The computation steps (Fig. 4) of the modified hybrid learning algorithm are described below.

(1) Population initialization using real-code: randomly create an initial population (Pop) of the premise parameters of Layer 1.

(2) Forward propagation from Layer 1 to Layer 4: utilize LSE for optimizing the set of consequent parameters.

(3) Solution evaluation and save best solution: fix the optimal consequent parameters and evaluate solutions by computing the following objective (or error) function.

$$f(s) = \frac{1}{2}\sum_{i=1}^{N}(e_i)^2 = \frac{1}{2}\sum_{i=1}^{N}\big(Q_f(i) - Q_o(i)\big)^2 \qquad (3)$$

where $f(s)$ is the objective function corresponding to the solution (s) of premise parameters. $e_i$ is the error of the $i$th data. N is the number of data. $Q_f(i)$ and $Q_o(i)$ are the forecasted and observed values of the $i$th data.

(4) Genetic operator procedure: (a) the reproduction procedure makes a duplicate of parent chromosomes as a tentative new population. The selection probability of the possible chromosomes for





the next generation is in proportion to the fitness value of the chromosomes. The selection procedure follows the schema theorem proposed by Goldberg (1989), i.e. the best chromosomes gains more duplicates while the worst ones are discarded. The tournament selection (Goldberg and Deb, 1991) is used in this study. (b) the crossover procedure with probability $P_c$ re-combines two parent chromosomes into new offspring chromosomes. (c) To maintain genetic diversity in the population, mutation procedure can be implemented occasionally with probability $P_m$ for the next generation.

(5) Stop criteria: Evaluate the created solutions through Steps 2-3. If the iteration number is less than the max generation "$G_{max}$", then repeat Steps 2-5. Otherwise, stop and output the optimization results.

The differences between the original and modified hybrid learning algorithms are: (1) the former fuses the LSE with steepest descent algorithm (Jiang, 1993; Zhang, 1999), while the latter fuses the LSE with GA for optimizing ANFIS parameters; (2) the former is deterministic and derivative optimization algorithm (Tamura et al., 2008), while the latter is stochastic and derivative-free optimization algorithm; (3) the former is local search technique and limited to the initial values of premise parameters of Layer 1, while the latter is global search technique and one of the state-of-the-art artificial intelligent techniques (Juang, 2002). The parameters of the steepest descent algorithm in original hybrid learning algorithm contain learning rate (Ita, $0 < \eta \leq 0.1$), momentum term (Alpha, $0 < \alpha \leq 1.0$) and max generation ($G_{max}$), while the parameters of the GA in modified hybrid learning algorithm contain population (Pop), max generation ($G_{max}$), crossover probability ($P_c$) and mutation probability ($P_m$).

**3.3 Evaluation criteria**

Considering the stochastic nature of hydrological variable, one must not rely on single criteria when evaluating the performance of data-driven models (ex. Cheng et al., 2017). In this paper, both visual plots in conjunction with statistical metrics are used to evaluate the models' performances. The Mean Absolute Error (MAE), Root Mean Square Error (RMSE), Coefficient of Efficiency (CE) and Coefficient of Correlation (CC) are selected to evaluate the forecasting accuracy of the three models. We also propose a coherent set of evaluation criteria to fully distill the robustness (reliability, vulnerability and resilience) of model. The evaluation criteria are described below.

(1) MAE

$$\text{MAE} = \frac{1}{N}\sum_{i=1}^{N}|Q_f(i) - Q_o(i)|, \quad \text{MAE} \geq 0 \tag{4}$$





(2) RMSE

$$\text{RMSE} = \sqrt{\frac{1}{N}\sum_{i=1}^{N}\big(Q_f(i) - Q_o(i)\big)^2}, \ \text{RMSE} \geq 0 \tag{5}$$

(3) CE

$$\text{CE} = 1 - \frac{\sum_{i=1}^{N}\big(Q_f(i) - Q_o(i)\big)^2}{\sum_{i=1}^{N}\big(Q_o(i) - \bar{Q}_o\big)^2}, \ \text{CE} \leq 1 \tag{6}$$

where $\bar{Q}_o$ is the average of the observed data.

(4) CC

$$\text{CC} = \frac{\sum_{i=1}^{N}(Q_f(i) - \bar{Q}_f)(Q_o(i) - \bar{Q}_o)}{\sqrt{\sum_{i=1}^{N}(Q_f(i) - \bar{Q}_f)^2 \sum_{i=1}^{N}(Q_o(i) - \bar{Q}_o)^2}}, \ -1 \leq \text{CC} \leq 1 \tag{7}$$

where $\bar{Q}_f$ is the average of the forecasted data.

(5) Reliability of model

This indicator can be described by the probability that model forecast remains in a satisfactory state.

$$\text{Reliability} = \frac{\sum_{i=1}^{N} K_i}{N} \times 100\% \tag{8a}$$

$$K_i = \begin{cases} 1, & \text{if}( \ \text{RAE}_i \leq \delta) \\ 0, & \text{else} \end{cases} \tag{8b}$$

$$\text{RAE}_i = \frac{|Q_f(i) - Q_o(i)|}{Q_o(i)} \times 100\%, \ \text{RAE}_i \geq 0 \tag{8c}$$

where $\text{RAE}_i$ is the Relative Absolute Error (RAE) of the $i$th data. $K_i$ is the number of times that RAE is less than or equal to the threshold value ($\delta$) of qualified forecast. The $\delta$ is set to 20% according to Chinese standard (GB/T 22482-2008).

(6) Vulnerability of model

Vulnerability represents the incompetence of a model to resist the effects of a hostile environment (e.g., the stochastic nature of hydrological variable). Vulnerability of model denotes the maximum RAE of model forecast.

$$\text{Vulnerability} = \max_{i=1}^{N}\{\text{RAE}_i\} \tag{9}$$

(7) Resilience of model

Resilience of model describes how quickly model forecast is likely to recover once unqualified forecast has occurred.





$$\text{Resilience} = \begin{cases} 100\%, & \text{if(Reliability} = 100\% \text{ or Vulnerability} \leq \delta) \\ \frac{\sum_{i=1}^{N-1} R_i}{N - \sum_{i=1}^{N} K_i} \times 100\%, & \text{else} \end{cases} \quad (10a)$$

$$R_i = \begin{cases} 1, & if ( \text{ RAE}_i > \delta \text{ and } \text{RAE}_{i+1} \leq \delta) \\ 0, & else \end{cases} \quad (10b)$$

where $R_i$ is the number of times that model forecast is likely to transfer from unqualified into qualified forecast in the $i$th data. Especially, the resilience is equal to 100% if the reliability is equal to 100% or vulnerability is less than the δ.

## 4 Results and discussion

### 4.1 Input selection and ANFIS model parameters

This study investigates the multi-step-ahead flood forecast performances of the three types of ANFIS models based on a large number of rainfall-runoff patterns of TGR in the Yangtze River. The flood forecast of TGR is a multi-inputs and single-output (MISO) pattern (Fig. 1) and there are 6,832 normalized input data (14 year*488 data/year) (Table 1). There are 12 input variables (10 flow variables and 2 regional rainfall variables) and their time-lag information is roughly estimated according to the geographical distribution and the various travel times of the flow from the upper stream and region rainfall gauged stations to the TGR (Table 2). The Gamma Test (Koncar 1997) as one of the state-of-the-art input selection techniques is used to identify the non-linearity of the rainfall-runoff relationship (ex. Chang & Tsai, 2016) and determine the optimal input combination of the three ANFIS models, as shown in Table 2. The input combinations of current variables Q(t) and R(t) have been selected as benchmarks, Ben(I) and Ben(II) for horizons t+8 and t+12, respectively. We only give the optimal (first) and suboptimal (second) input combinations at horizons t+4 (one-day), t+8 (two-day) and t+12 (three-day). We find that: (1) the combination associated with the lowest *Ratio* values (0.0001, 0.0012 and 0.0015) of Gamma Test are considered as the best combination at horizons t+4, t+8 and t+12, respectively; (2) the rainfall variables R(t-2) and R(t) in I Region are selected as optimal time-lagged rainfall variables at horizons t+4 and t+8, respectively; (3) the flow variable Q(t) in Fuxi stream gauged station is not necessary to form the best combination at horizon t+12. This implies that some of input variables at horizons t+8 and t+12 are not necessary to form the best combination as they may make no




contribution for rainfall-runoff pattern, while all input variables at horizon t+4 are necessary to form the best combination as they make crucial contribution for rainfall-runoff pattern.

For inflow of TGR, flood forecast models at horizons t+4, t+8 and t+12 are constructed by the C-ANFIS (Model 1), R-ANFIS (Model 2) and MR-ANFIS (Model 3), and there are 4,392 normalized training dataset (9 year*488 data/year) and 2,440 normalized testing dataset (5 year*488 data/year) (Table 1). The parameters in every ANFIS model contain: (a) the number of input variables ($N_1$) at horizons t+4, t+8 and t+12 are 12, 7 and 4, respectively (Table 2); (b) the number of output variables ($N_2$) is 1 (Single-output pattern); (c) the number of membership functions ($N_3$) for every input variable is 2 ($N_3$=Clusters); (d) the number of premise parameters ($N_4$) in Layer 1 at horizons t+4, t+8 and t+12 are 48($2\times12\times2$), 28($2\times7\times2$) and 16($2\times4\times2$), respectively ($N_4=2\times N_1 \times N_3$); (e) the number of consequent parameters ($N_5$) in Layer 4 at horizons t+4, t+8 and t+12 are 26=$2\times(12+1)$, 16=$2\times(7+1)$ and 10=$2\times(4+1)$, respectively ($N_5=N_3\times(N_1+N_2)$).

After implementing an intensive trial-and-error procedure based on the training data set, every ANFIS model is constructed to have five layers with two membership functions for every input variable, which in general would have the most suitable performances for the three models of TGR. The ANFIS architectures in the three models are then applied to the testing data set without further modifications. In the Models 1 & 2, the parameters of the steepest descent algorithm in the original hybrid learning algorithm (LSE & Steepest descent algorithm) are set as $\eta = 0.01$, $\alpha = 0.9$ and $G_{max}$=1000. In the Model 3, the parameters of the GA in the modified hybrid learning algorithm (LSE & GA) are set as Pop=1,000, $G_{max}$=1,000, $P_c$ =0.9 and $P_m$=0.1. The summarized results of the three ANFIS models are presented below.

### 4.2 Fuzzy rules and performance of ANFIS models

As mentioned earlier, the fuzzy if-then rules used in the ANFIS models are of the Sugeno's type. Taking the Model 3 at horizon t+4 as an example, the inputs consist of three types (10 upstream flow variables, 2 regional rainfall variables and TGR flow variable). The Model 3 has two clusters ($N_3$), and its Gaussian membership functions of the inputs (XJB reservoir flow, I regional rainfall and TGR flow) at horizon t+4 are shown in Fig. 5. Only the membership function at horizon t+4 is shown because the membership functions at horizons t+8 and t+12 are similar to each other. For the membership functions





of TGR flow, it is clear that cluster 1 represents low flow while cluster 2 represents high flow. The membership functions for lagged tributaries flows and region rainfalls are also associated with the two clusters that represent higher input amount and lower input amount (Fig. 5).

The values of MAE, RMSE, CE, CC, Reliability, Vulnerability and Resilience for each models are summarized in Table 3. The results show that: (1) the MAE, RMSE, Vulnerability values of Model 3 are relative smaller than the other two models at all horizons; (2) the CE, CC, Reliability and Resilience values of Model 3 are relative larger than the other two models at all horizons. These values indicate the more accurate and robust results we might reach based on the recurrent learning mechanism and modified hybrid learning algorithm.

We further examine the results of Model 1 and Model 2, which have exactly the same optimization algorithm (the original hybrid learning algorithm) with the different learning mechanisms. The Model 1 & 2 utilize the static and dynamic (recurrent) learning mechanisms, respectively. Apparently, the Model 2 have much better performances, in terms of much smaller MAE, RMSE and Vulnerability values as well as larger CE, CC, Reliability and Resilience values, than Model 1 for both training and testing cases at horizons t+4, t+8 and t+12, especially at horizons t+8 and t+12. These results provide clear and rigid evidences that the accuracy and generalization of the constructed models are poor in the two or three days ahead flood forecast if the recurrent learning mechanism cannot be used in ANFIS.

And then, we check the results of Model 2 and Model 3, which have exactly the same learning mechanism (the recurrent learning mechanism) with the different hybrid learning algorithms. The Model 2 & 3 utilize the original and modified hybrid learning algorithms, respectively. As we compared the results of Model 2 and Model 3, the MAE, RMSE and vulnerability values of the Model 3 are smaller than that of the Model 2, and the CE, CC, reliability and resilience values of the Model 3 are larger than those of the Model 2 for both training and testing periods at all horizons. For instance, the reliabilities of horizon t+12 in the training period are 96.19% and 99.16% for the Models 2 & 3, respectively; and the reliabilities of horizon t+12 in the testing period are 92.15% and 96.80% for the Models 2 & 3, respectively. These results indicate that the Model 3 provides much better (accurate and robustness) forecasts than the Model 2.





The growth trends of the MAE, RMSE and vulnerability as well as the decline trends of the CE, CC, reliability and resilience values of the Model 1 & 2 at horizons t+4, t+8 and t+12 in the testing period are shown in Fig. 6(a). It is noticed that the Model 1 outputs would gradually apart from true values as the forecasting time-step increased, and using static learning mechanism and constant synaptic weights would further accelerate the growth of forecast errors. Whereas the Model 2 outputs would be close to true values as the forecasting proceeds, using recurrent learning mechanism and updating synaptic weights would correct forecast errors in real time. The MAE, RMSE, vulnerability values of Model 2 increase gradually, whereas those of Model 1 increase rapidly from the horizon t+4 to t+12. The CE, CC, reliability and resilience values of the Model 2 decrease slowly, while those of Model 1 decrease quickly from the horizon t+4 to t+12. The two models perform equally well for horizon t+4 forecasting, while significant differences among their performances are found as the forecasting horizons from t+8 to t+12. This demonstrates that the Model 2 has substantially smaller error accumulation and propagation than the Model 1, and the Model 2 could provide reasonable results for multi-step-ahead flood forecast if real-time observed rainfall and runoff as well as the feedback of model runoff outputs to the input layer and consequent layer can be implemented as the forecasting proceeds.

Fig. 6(b) also presents the radar maps of MAE, RMSE, CE, CC, reliability, vulnerability and resilience values of the Models 2 & 3 at horizon t+4, t+8 and t+12 in the testing period. It clearly indicates that the Model 3 produces much lower MAE, RMSE and vulnerability and higher CE, CC, reliability and resilience values than Model 2. Such result is consistent with the parameter optimization algorithm affecting the robustness of recurrent ANFIS model, which is determined by the modified hybrid learning algorithm addressed in Section 3.2.

Fig. 7 shows the residual values of the three models at horizons t+4, t+8 and t+12 in testing period. The Model 3 produces the fewer number of residual values falling outside the ± 20% range of observed values than the other two models, while the Model 1 performs even worse than the Model 2. Meanwhile, the Model 3 produces smaller maximum residual value than the other two models, while the Model 1 performs even worse than the Model 2. Such result implies that the recurrent learning mechanism and parameter optimization algorithm (LSE & GA) could reduce the highly variable



rainfall-runoff information and thus increase the reliability of the ANFIS.

To illustrate the forecasting accuracy of the three models, the scatter plots of the forecasted and observed values of the three models at horizons t+4, t+8 and t+12 in the testing period are shown in Fig. 8. In the Model 3, almost all pairs of forecasted and observed points at horizons t+4, t+8 and t+12 scatter closely to the diagonal line for TGR inflow. In the Model 2, most of pairs of forecasted and observed points at horizons t+4, t+8 and t+12 scatter closely to the diagonal line for TGR inflow. In the Model 1, only the pairs of forecasted and observed points at horizons t+4 scatter suitably around the diagonal line for TGR inflow. According to the results, the Model 3 provides a significant superior performance to the Model 1 and a comparative superior performance to the Model 2.

Taylor diagram provides a way of graphically summarizing how closely a pattern (or a set of patterns) matches observations (Taylor, 2001). The similarity between two patterns is quantified in terms of three statistic indicators, i.e., correlation, centered RMSE and standard deviation. Such diagram is especially useful in evaluating the relative skill of many different models. Taylor diagram (Fig. 9) systematically shows these statistic indicators of the three models at horizons t+4, t+8 and t+12 in the testing period. Taking the horizon t+12 for an example, Fig. 9 easily identifies Model 3 that performs relative well (CC = 0.97, standard deviation = 9022 cms & centered RMSE = 2206 cms) because it lies relative close to the observed point (CC = 1.0 & standard deviation = 8991 cms) than the Model 2 (CC = 0.95, standard deviation = 9060 cms & centered RMSE = 2855 cms) and Model 1 (CC = 0.81, standard deviation = 9085 cms & centered RMSE = 5572 cms).

## 5 Conclusions

A robust recurrent ANFIS for modeling multi-step-ahead flood forecast of the Three Gorges Reservoir is presented in this study. A modified hybrid learning algorithm which fuses the LSE with GA is used to optimize the parameters of recurrent ANFIS. This method is compared with the original hybrid learning algorithm which fuses LSE with steepest descent algorithm. The classic ANFIS is selected as benchmark, and its results are compared with those of recurrent ANFIS for the purpose of evaluating the performance of recurrent (dynamic) learning mechanism in multi-step-ahead flood forecast. The comparison between recurrent ANFIS and modified recurrent ANFIS focuses on how the different




hybrid learning algorithms affect the robustness of recurrent ANFIS models. Three models with their optimal input variables identified by the Gamma Test are applied for modeling one to three days ahead flood forecasts of the Three Gorges Reservoir in the Yangtze River. A detail comparison of three models' performances has been given based on a coherent set of evaluation criteria, which are the MAE, RMSE, CE and CC for assessing the forecasting accuracy and the reliability, vulnerability and resilience for distilling the robustness.

The results showed that the recurrent ANFIS model can enhance the forecast accuracy and prolong forecast horizon in comparison to the classic ANFIS model. Moreover, the modified recurrent ANFIS model not only enhance the forecast accuracy, but also improve the robustness of recurrent ANFIS model. That is, the recurrent learning mechanism can improve the generalization of ANFIS model and the modified hybrid learning algorithm (LSE & GA) can overcome the shortcomings which are mainly instability, sensitivity to initial conditions, local minima problems of the original hybrid learning algorithm (LSE & steepest descent algorithm). Thus, the modified recurrent ANFIS model can successfully be implemented as a reliable and accurate alternative for multi-step-ahead flood forecast. Our results demonstrate the proposed model (MR-ANFIS) can make accurate and robust multi-step-ahead flood forecasts in the Three Gorges Reservoir, which would provide precious decision-making time for effectively managing contingencies and emergencies and greatly alleviating flood risk as well as loss of life and property.

**Acknowledgements**

This study is financially supported by the National Key Research and Development Project of China (Grant No. 2016YFC0402206), the Ministry of Science and Technology, Taiwan, ROC (MOST 104-2811-B-002-146) and the National Natural Science Foundation of China (Grant No. 41401018).

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





Table 1 TGR inflow datasets in flood season (from June 1st to September 30th and time step = 6h)

| Period | Year | Max (cms) | Ave (cms) | Min (cms) | Std Dev (cms) |
|---|---|---|---|---|---|
| Training | 2003 | 50800 | 24894 | 9980 | 8472 |
| | 2004 | 67300 | 22270 | 11200 | 7844 |
| | 2005 | 54900 | 25680 | 12500 | 9373 |
| | 2006 | 31800 | 13564 | 6940 | 4444 |
| | 2007 | 56900 | 24583 | 8860 | 9301 |
| | 2008 | 40600 | 22876 | 10000 | 7114 |
| | 2009 | 57100 | 21880 | 8440 | 8731 |
| | 2010 | 68300 | 24470 | 9460 | 10806 |
| | 2011 | 45400 | 17759 | 8180 | 7550 |
| Testing | 2012 | 69100 | 26581 | 10500 | 12199 |
| | 2013 | 49000 | 21384 | 10500 | 8071 |
| | 2014 | 55000 | 24951 | 11000 | 8574 |
| | 2015 | 39000 | 19298 | 10000 | 5550 |
| | 2016 | 50000 | 20262 | 10000 | 7036 |

Table 2 The optimal lagged variables and input combinations used in the three models

| Station | Travel time | Sub-opt | Opt | Ben (I) | Sub-opt | Opt | Ben (II) | Sub-opt | Opt |
|---|---|---|---|---|---|---|---|---|---|
| Xiangjiaba | 48h | Q(t-4) | Q(t-4) | Q(t) | Q(t) | Q(t) | Q(t) | Q(t) | Q(t) |
| Hengjiang | 48h | Q(t-4) | Q(t-4) | Q(t) | Q(t) | Q(t) | Q(t) | Q(t) | Q(t) |
| Fuxi | 42h | Q(t-3) | Q(t-3) | Q(t) | Q(t) | Q(t) | Q(t) | Q(t) | / |
| Gaochang | 48h | Q(t-4) | Q(t-4) | Q(t) | Q(t) | Q(t) | Q(t) | Q(t) | Q(t) |
| Fushun | 42h | Q(t-3) | Q(t-3) | Q(t) | Q(t) | Q(t) | Q(t) | / | / |
| Chishui | 24h | Q(t) | Q(t) | Q(t) | / | / | Q(t) | / | / |
| Wucha | 12-18h | Q(t) | Q(t) | Q(t) | / | / | Q(t) | / | / |
| Beibei | 12-18h | Q(t) | Q(t) | Q(t) | / | / | Q(t) | / | / |
| Wulong | 6-12h | Q(t) | Q(t) | Q(t) | / | / | Q(t) | / | / |
| I Rainfall | 42-48h | R(t-1) | R(t-2) | R(t) | / | R(t) | R(t) | / | / |
| II Rainfall | 12-18h | R(t) | R(t) | R(t) | / | / | R(t) | / | / |
| TGR | / | Q(t) | Q(t) | Q(t) | Q(t) | Q(t) | Q(t) | Q(t) | Q(t) |
| Horizon | / | t+4 | t+4 | t+8 | t+8 | t+8 | t+12 | t+12 | t+12 |
| *Ratio* | / | 0.0007 | 0.0001 | 0.0115 | 0.0081 | 0.0012 | 0.0121 | 0.0097 | 0.0015 |



Table 3 Performance of the three models for TGR flood forecast at horizons t+4, t+8 and t+12

| Model | | Model 1 | | | Model 2 | | | Model 3 | | | Model 2 VS 1 (%) | | | Model 3 VS 2 (%) | | |
|---|---|---|---|---|---|---|---|---|---|---|---|---|---|---|---|---|
| | Horizon | t+4 | t+8 | t+12 | t+4 | t+8 | t+12 | t+4 | t+8 | t+12 | t+4 | t+8 | t+12 | t+4 | t+8 | t+12 |
| MAE (cms) | Training | 1021 | 1635 | 3006 | 713 | 803 | 842 | 679 | 765 | 787 | -30.17 | -50.87 | -71.99 | -4.76 | -4.76 | -6.54 |
| | Testing | 1413 | 1990 | 3722 | 899 | 1013 | 1190 | 832 | 921 | 1072 | -36.41 | -49.09 | -68.03 | -7.41 | -9.09 | -9.91 |
| RMSE (cms) | Training | 1712 | 2462 | 4199 | 1131 | 1306 | 1319 | 992 | 1146 | 1157 | -33.94 | -46.94 | -68.59 | -12.28 | -12.28 | -12.28 |
| | Testing | 2208 | 2896 | 5410 | 1596 | 2351 | 2814 | 1388 | 2062 | 2426 | -27.71 | -18.83 | -47.98 | -13.04 | -12.28 | -13.79 |
| CE | Training | 0.95 | 0.91 | 0.75 | 0.98 | 0.97 | 0.97 | 0.99 | 0.98 | 0.98 | 3.17 | 6.62 | 29.36 | 1.01 | 1.01 | 1.01 |
| | Testing | 0.92 | 0.88 | 0.66 | 0.95 | 0.92 | 0.91 | 0.97 | 0.94 | 0.93 | 3.33 | 4.68 | 38.09 | 2.04 | 2.04 | 2.04 |
| CC | Training | 0.97 | 0.95 | 0.88 | 0.98 | 0.98 | 0.98 | 0.99 | 0.99 | 0.99 | 1.04 | 3.17 | 11.38 | 1.01 | 1.01 | 1.01 |
| | Testing | 0.96 | 0.93 | 0.81 | 0.96 | 0.95 | 0.95 | 0.98 | 0.97 | 0.97 | 0.04 | 2.22 | 17.36 | 2.04 | 2.04 | 2.04 |
| Reliability (%) | Training | 97.23 | 93.77 | 75.04 | 97.32 | 96.52 | 96.19 | 99.25 | 99.20 | 99.16 | 0.09 | 2.94 | 28.18 | 1.98 | 2.77 | 3.09 |
| | Testing | 96.10 | 92.65 | 72.43 | 96.92 | 94.19 | 92.15 | 99.07 | 98.11 | 96.80 | 0.86 | 1.66 | 27.23 | 2.22 | 4.17 | 5.05 |
| Vulnerability (%) | Training | 36.85 | 47.20 | 68.15 | 30.84 | 37.35 | 62.16 | 30.23 | 36.61 | 60.94 | -16.30 | -20.88 | -8.78 | -1.97 | -1.96 | -1.96 |
| | Testing | 40.29 | 61.45 | 75.13 | 32.79 | 41.01 | 68.13 | 30.15 | 37.53 | 61.85 | -18.62 | -33.26 | -9.33 | -8.05 | -8.47 | -9.22 |
| Resilience (%) | Training | 48.86 | 43.66 | 30.55 | 77.87 | 75.31 | 69.58 | 78.66 | 77.08 | 71.29 | 59.38 | 72.49 | 127.75 | 1.01 | 2.35 | 2.46 |
| | Testing | 40.08 | 37.89 | 27.41 | 75.06 | 68.14 | 64.86 | 76.90 | 70.55 | 67.20 | 87.28 | 79.83 | 136.61 | 2.45 | 3.54 | 3.61 |

Notes: in comparison analysis between models, the negative value denotes decrease and positive value denotes increase.





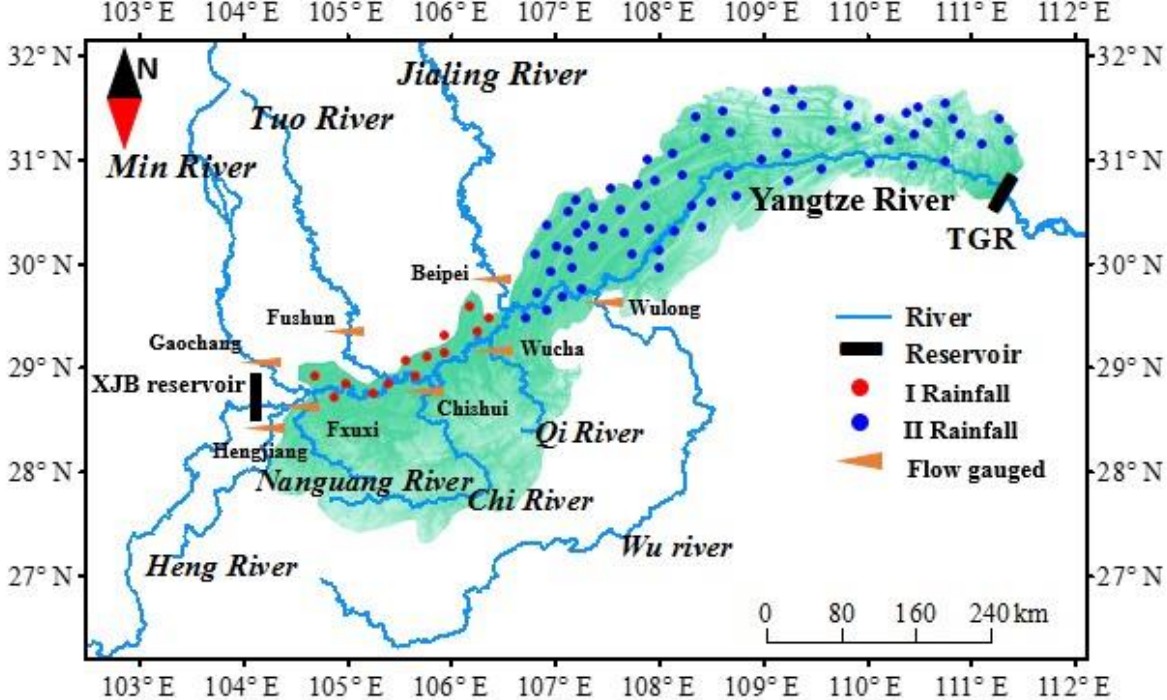

Figure 1 Map of the Yangtze River basin between the Xiangjiaba (XJB) and TGR reservoirs





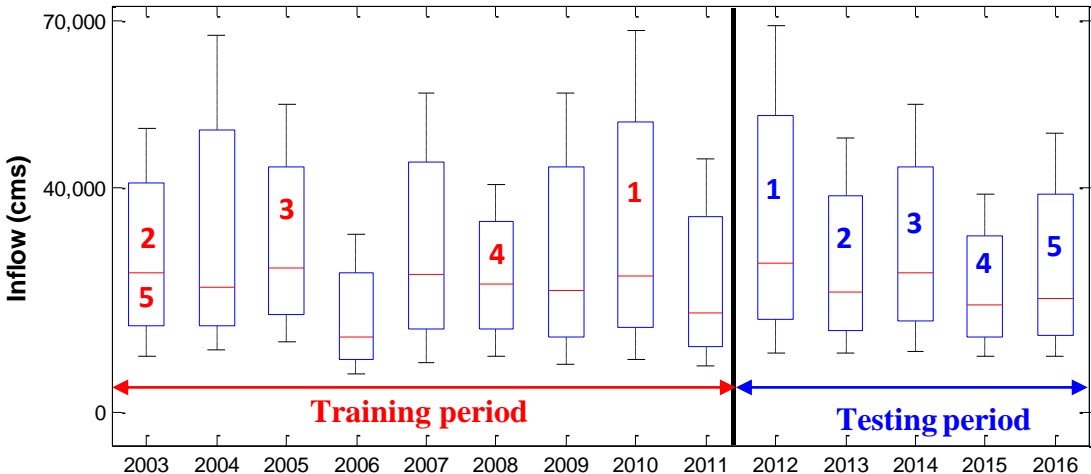

Figure 2 The boxplot of the observed TGR inflows in the flood season (June 1st to September 30th) from 2003 to 2016 year (14 years) with 6 hours' time-step





Figure 3 Framework of three ANFIS modeling approaches



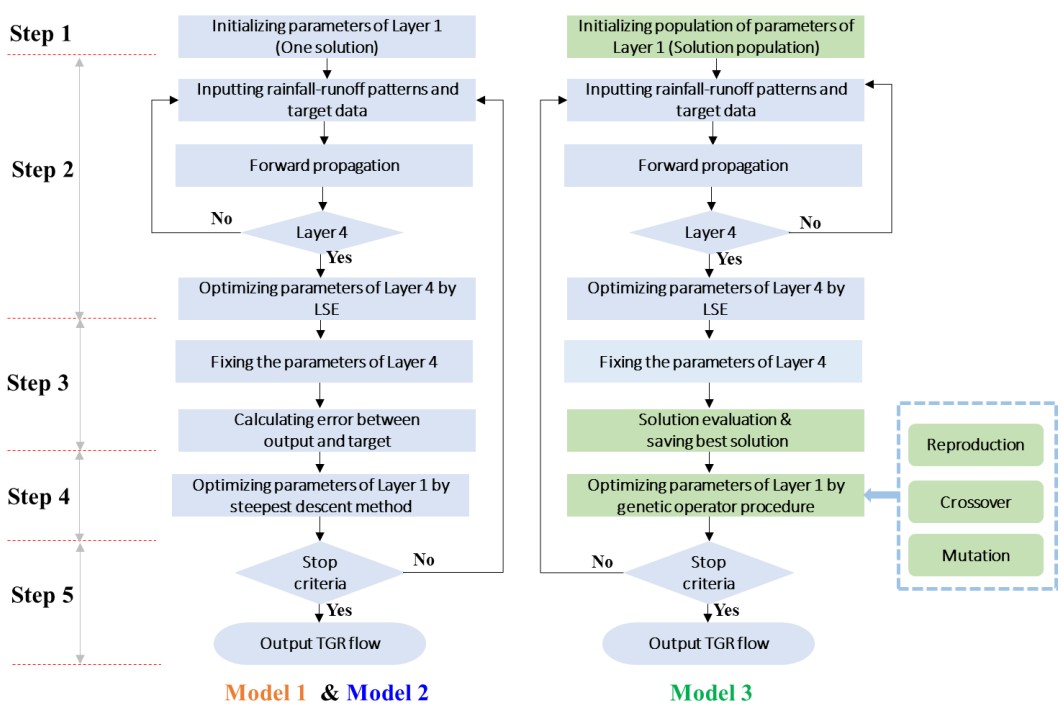

Figure 4 Computation steps of the original (left) and modified (right) hybrid learning algorithms





Figure 5 Membership functions of the inputs (XJB flow, I regional rainfall and TGR flow variables) in Model 3 at horizon t+4





**Figure 6** The trend plots (left) and radar maps (right) of evaluation indicator values of the Model 2 & 3 at horizons t+4, t+8 and t+12 in the testing period



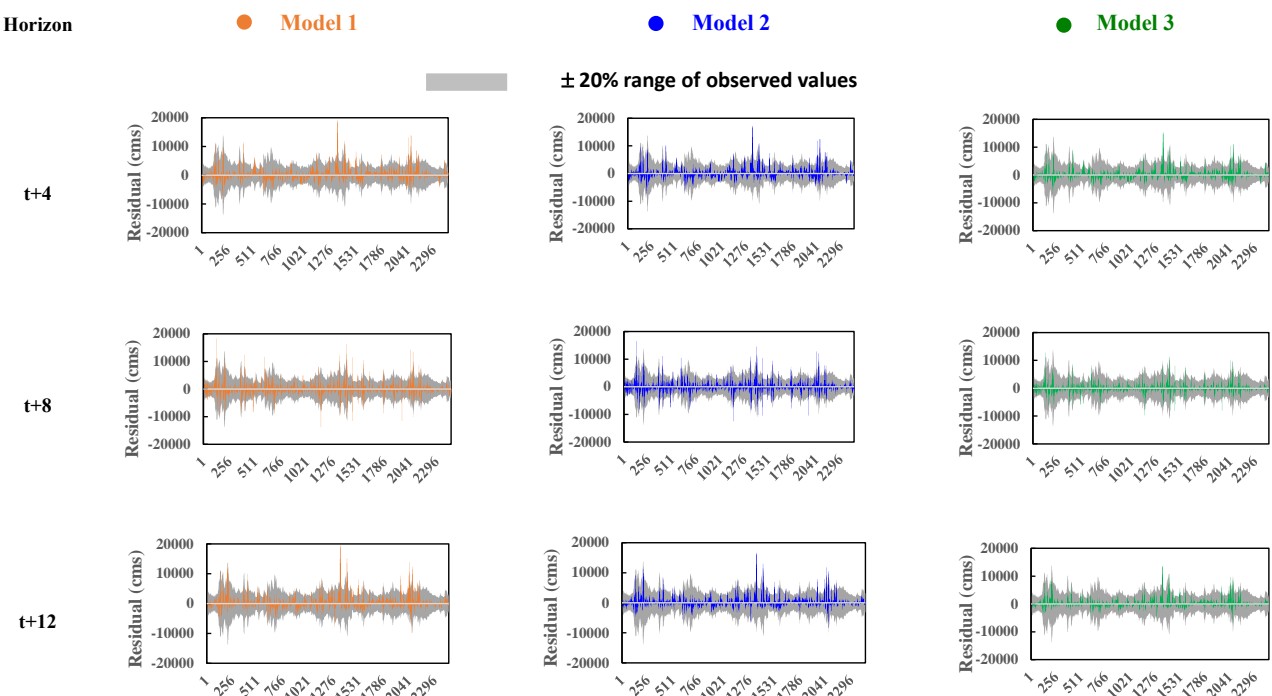

Figure 7 The residual values (=Observation - Forecast) of the three models at horizons t+4, t+8 and t+12 in testing period





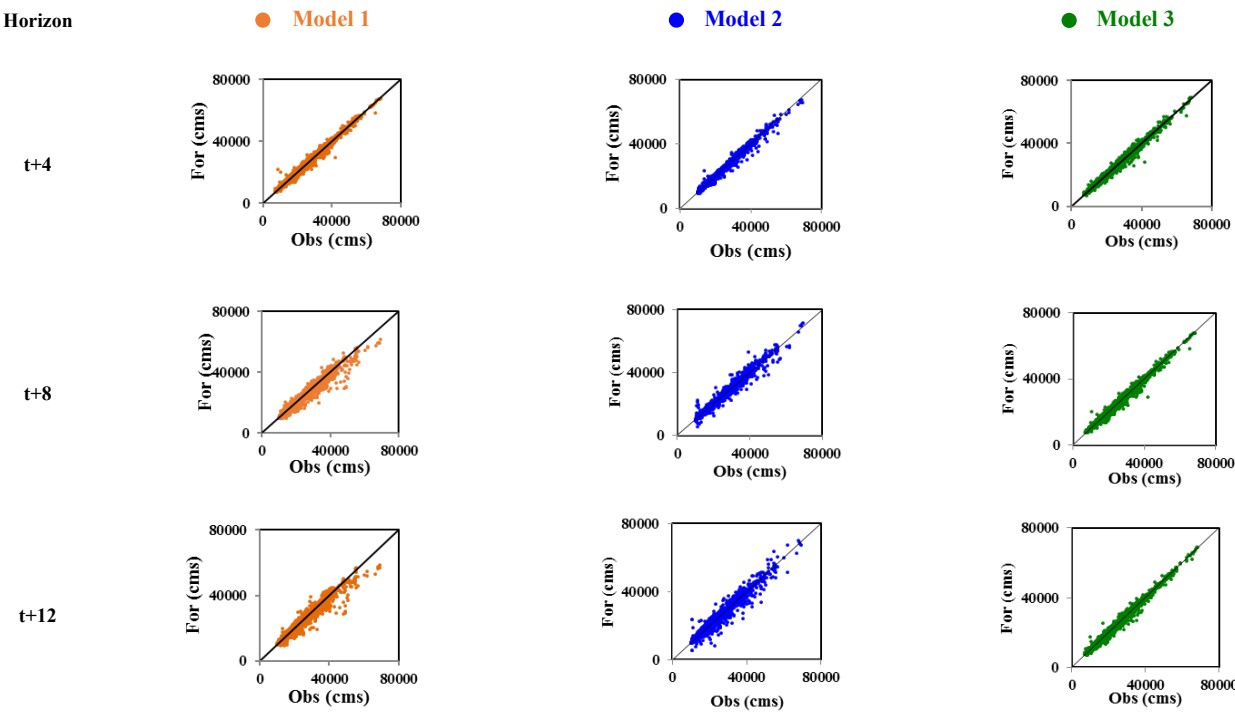

Figure 8 Scatter plots of the forecasted (For) and observed (Obs) values of reservoir inflow at horizons t+4, t+8 and t+12 in the testing period





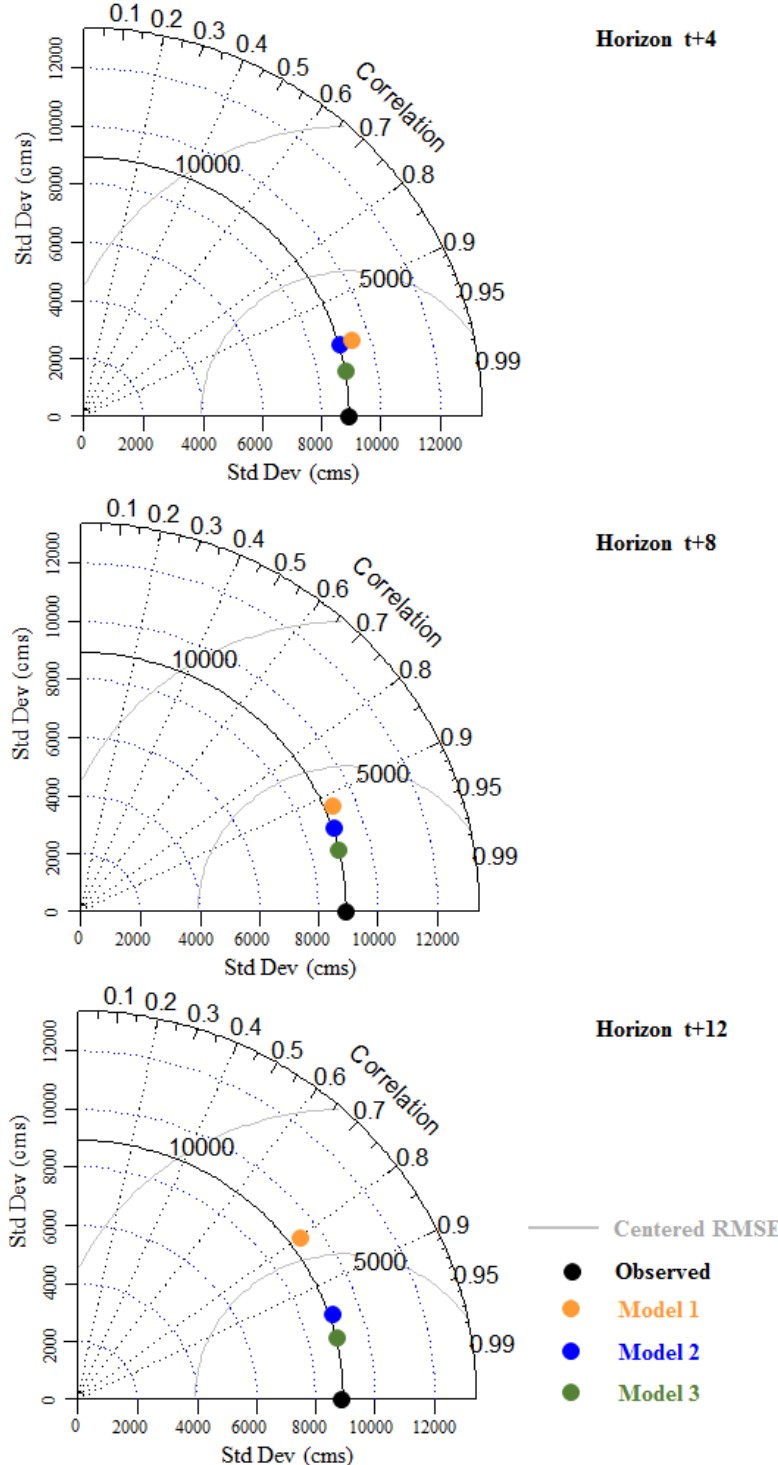

Figure 9 Taylor diagrams of the three models at horizons t+4, t+8 and t+12 in the testing period