# Peer review of "A robust recurrent ANFIS for modeling multi-step-ahead flood forecast of Three Gorges Reservoir in the Yangtze River"

_Hydrology and Earth System Sciences, 2017_

## Referee Comment (RC1) · Anonymous Referee #1 · 25 Sep 2017

The paper is well written. The significance of the result is not very high, since in some cases only a small improvement is obtained by using the proposed approach with respect to existing approaches. However, the paper is technically correct. Some suggestions to improve the paper follow.

1. There is a typo at line 143 (Fig. 2 should be Fig. 3)

2. actually, all the part from line 143 to line 157 is standard and can be replaced by a very simple description of the well-known structure of the ANFIS. (actually, this is an issue for the Associate Editor handling the submission. If she/he thinks that the audience of the journal is acquainted with fuzzy systems, then he can suggest to

remove the part, otherwise, keep as it is).

---

## Author Comment (AC1) · 2 Oct 2017

Dear Professor Roberto Greco,

We are very glad to learn reviewers' recognition on the value of our study and sincerely thank the constructive comments and suggestions to our manuscript (Ref: hess-2017-457). This revision focused on highlighting the paper's contribution and originality. Responses were made to each and every comment raised by the reviewer, and revisions were incorporated into the revised manuscript. Changes made in the revised manuscript were colored in blue. Modifications in the revised manuscript were made to meet the publication standards required by the HESS.

[Figure]

As known, accurate and robust multi-step-ahead flood forecast during flood season is extremely crucial to reservoir flood control. A modified hybrid learning algorithm, which fuses the Least Square Estimator (LSE) with Genetic Algorithm (GA), is proposed for optimizing the parameters of recurrent ANFIS (R-ANFIS) model to overcome the instability and local minima problems as well as improve model's generalization and robustness. We wish to have an opportunity to share our research methodology and findings with readers through the HESS.

Best Regards,

Fi-John Chang Professor, Department of Bioenvironmental Systems Engineering.

Yanlai Zhou Post-doctor, Department of Bioenvironmental Systems Engineering.

National Taiwan University.

Email: changfj@ntu.edu.tw (F. Chang); zyl23bulls@whu.edu.cn (Y. Zhou).

Tel: +886-2-33663452.

Responses to Referee #1: The paper is well written. The significance of the result is not very high, since in some cases only a small improvement is obtained by using the proposed approach with respect to existing approaches. However, the paper is technically correct.

Reply: We deeply appreciate you for your encouragement and recognition on the value of our study. The constructive comment is sincerely appreciated. We have added the following statements and Fig. 10 to enhance our presentation regarding the significance and reliability of the results obtained by our proposed models (M-ANFIS).

To demonstrate the significance and effectiveness of the proposed methodology that fuses the LSE with Genetic Algorithm (GA) for optimizing the parameters of R-ANFIS. Fig. 10 further shows sensitive analysis of the three models at horizon t+4, t+8 and t+12 in the testing period with cross-validation (exchanging one-year dataset between

training and testing periods, see Fig. 2). Fig. 10 clear indicates that the forecast accuracy and robustness of M-ANFIS (Model 3) are significantly superior to the others (Model 1&2), as all of the indicators (MAE, RMSE, CE, CC, Reliability, Vulnerability and Resilience) in Model 3 have smaller boxplots and are insensitive to cross-validation process at different horizons. This analysis suggests that there is great capable for overcoming the local minima and instability problems of the original hybrid learning algorithm (coupling steepest descent algorithm with Least Square Estimator) as the proposed methodology is implemented.

Some suggestions to improve the paper follow.

1. There is a typo at line 143 (Fig. 2 should be Fig. 3).

Reply: We thank you for taking time to read our manuscript and give valuable comments for it. Sorry for such mistake. We have changed Fig.2 into Fig.3.

2. Actually, all the part from line 143 to line 157 is standard and can be replaced by a very simple description of the well-known structure of the ANFIS. (actually, this is an issue for the Associate Editor handling the submission. If she/he thinks that the audience of the journal is acquainted with fuzzy systems, then he can suggest to remove the part, otherwise, keep as it is).

Reply: Thank you for your valuable comment. We agree and replace the part (Lines 143-157) by a simple description as follow:

The ANFIS consist of five layers (see Fig.3). The Gaussian function and linear function are selected as the member function for Layer 1 and consequent function for Layer 4, respectively. The T-norm operator is applied to fuzzy-AND operation in Layer 2. For a full description of the well-known structure of the ANFIS, please refer to Jang (1993).

The authors would like to thank the editor and anonymous reviewers for their review and valuable comments related to this manuscript.
Please also note the supplement to this comment:
https://www.hydrol-earth-syst-sci-discuss.net/hess-2017-457/hess-2017-457-AC1-supplement.pdf

—————————————————————

[revised manuscript text omitted]
 with cross-validation (exchanging one-year dataset between training and testing periods, see Fig. 2). Fig. 10 clear indicates that the forecast accuracy and robustness of M-ANFIS (Model 3) are significantly superior to the others (Model 1&2), as all of the indicators (MAE, RMSE, CE, CC, Reliability, Vulnerability and Resilience) in Model 3 have smaller boxplots and are insensitive to cross-validation process at different horizons. This analysis suggests that there is great capable for overcoming the local minima and instability problems of the original hybrid learning algorithm (coupling steepest descent algorithm with Least Square Estimator) as the proposed methodology is implemented.

**5 Conclusions**

A robust recurrent ANFIS for modeling multi-step-ahead flood forecast of the Three Gorges Reservoir is presented in this study. A modified hybrid learning algorithm which fuses the LSE with GA is used to optimize the parameters of recurrent ANFIS. This method is compared with the original hybrid learning algorithm which fuses LSE with steepest descent algorithm. The classic ANFIS is selected as benchmark, and its results are compared with those of recurrent ANFIS for the purpose of evaluating

the performance of recurrent (dynamic) learning mechanism in multi-step-ahead flood forecast. The comparison between recurrent ANFIS and modified recurrent ANFIS focuses on how the different hybrid learning algorithms affect the robustness of recurrent ANFIS models. Three models with their optimal input variables identified by the Gamma Test are applied for modeling one to three days ahead flood forecasts of the Three Gorges Reservoir in the Yangtze River. A detail comparison of three models' performances has been given based on a coherent set of evaluation criteria, which are the MAE, RMSE, CE and CC for assessing the forecasting accuracy and the reliability, vulnerability and resilience for distilling the robustness.

The results showed that the recurrent ANFIS model can enhance the forecast accuracy and prolong forecast horizon in comparison to the classic ANFIS model. Moreover, the modified recurrent ANFIS model not only enhance the forecast accuracy, but also improve the robustness of recurrent ANFIS model. All of the indicators (MAE, RMSE, CE, CC, reliability, vulnerability and resilience) in MR-ANFIS model have smaller boxplots and are insensitive to cross-validation process at different horizons. 
[revised manuscript text omitted]

---

## Referee Comment (RC2) · Anonymous Referee #2 · 3 Oct 2017

Review of the paper hess-2017-457: A robust recurrent ANFIS for Modeling multi - Step Ahead flood forecast of Three Gorges Reservoir in the Yangtze River

General comments

The paper deals with the forecasting of the flood of the TGR using ANFIS model with three versions of this solution. Generally the paper is not extremely rigorous. For example the goal is to predict the flood of the dam. But what is the flood of the dam: the flood of the input river (how long upstream to not being influenced by the level in the reservoir)? The flood at the output of the reservoir (to be able to managed flood downstream)?

[Figure]

For the same lack of accuracy, the following sentence has no meaning, mathematically speaking "Vulnerability represents the incompetence of a model to resist the effects of a hostile environment (e.g., the stochastic nature of hydrological variable". It should be better to correct this sentence and to be more accurate and more mathematical in several occasions in the paper (please, see technical comments). Moreover, in my opinion being stochastic is not the real problem. The real problem comes from the ungaussian property of hydrologic signals.

Suffering from the same cause, it is not so easy to know if the models are recurrent or no recurrent. This question is essential because the design of a really recurrent model, where the input is the previously calculated output, is more difficult. It seems to appear that the models are not recurrent (the input seems to be the previous measured discharge plus upstream measured discharges and rains, see table 2). If the model uses previous observed discharge (using $Qo(t)$ to forecast $Qf(t+i)$) then it is mandatory to evaluate the quality using the persistency criteria in order to appreciate if the model has an added value or not (please see technical comments). Also, all along the paper the variable $Q(t)$ is used. Nevertheless the variables $Qf$ (forecasted discharge) or $Qo$ (observed discharge) are also defined. Thus what does $Q$ represent?

The procedure of training is not accurately described (p6 L13: "After implementing an intensive trial-and-error procedure). Let me recall that the paper must be sufficiently accurate to could be reproduced by other people. It is evident that it is not the case. P11, what is "the recurrent learning mechanism"? The same applies to the procedure of variable selection: the method is not described. But variable selection is essential in data driven models.

Finally, the section presenting results is quite confuse and difficult to read. Maybe some Tables and example of predicted signals, at each lead time, should be better to compare the models than the proposed indirect representations. Usually indirect representations (Fig 6, Fig 7) hide the defect of flood prediction when the peak is not good but the rest of the hydrograph quite well represented. For this reason it is

essential in case of flood prediction to provide an accurate measurement of the quality of the predicted peak, or a representation of the signals.

In conclusion, this lack of rigor must be corrected. The question about the kind of model (recurrent or not) must find a response. Only after this response it will be possible to evaluate the quality of the evaluation of the results. Flood forecasting is very difficult and I encourage the authors to deal with more accurately.

Specific comments

- Title: could you justify why the model is qualified of "robust" in the title?

- Abstract

It is not evident, reading only the abstract to know what are the criteria CC and CE, it is thus necessary to provide, at least, the name of the criteria in the abstract: for example Ce is the Nash-Sutcliff criterion or the coefficient of determination. And CC is the linear coefficient of correlation.

- Section 3.3. Evaluation criteria.

The aim of the paper is to provide prediction. Usually, in this case it is necessary to use a criteria specific to prediction, for example du persistency criteria (Kitanidis, P. K. and Bras, R. L.: Real-time forecasting with a conceptual hydrologic model: 2. Applications and results, Water Resour. Res., 16(6), 1034–1044, doi:10.1029/WR016i006p01034, 1980.). We suggest to authors to calculate also this criteria.

This criteria is mandatory when previous measured discharges are used to calculate future discharges, but it is no clear in the paper if previous observed discharges are used or only previous simulated discharges: having exact equations should remove the question. In table 2 it is not so clearly indicated if the Q(t) of TGR refers to observed or simulated discharge (Qf or Qo)? If it is Qo, then the model is not recurrent at al. The model can simulate a dynamic basin but it is static (finite impulse response). To verify if the model has a utility it is also possible to calculate the Nash criterion of the signal

[Figure]

Qo(t+lag). If the Nash criterion of the prediction Qf(t+lag) has a better Nash criterion than the previous one (on Qo), then the predictor is useful; in the contrary case, the model has no interest at all, it is only a model that duplicate, at its output, the received input. This behavior is easy to detect when predicted signals are provided, but it is not the case in this paper. This is a shame.

Technical corrections

P5L15: correct in Fig 3.

Notations in eq 3 are nor fully coherent: i, which is the number of a considered example, appears sometimes in index, sometime in parenthesis. P6L1-2 parameters are not linear or nonlinear. They are used in a linear combination or in a nonlinear function.

P6L8: it is necessary to add the equation of the 3 models to express clearly the inputs and outputs variables of the models. Unhopefully there is a great confusion in the literature about the concept of recurrent (infinite impulse response) and static (Finite impulse response). Could you add the equations?

Eq 9 the criteria RAE is not so good because it could be very high in case of low discharge. It is thus not good when there is very low and very high discharges? In the case of the 3 rivers it is not possible to have our own idea as signals are not provided. In P9 and others, please used accurate notation: not Q but Qf or Qo. To be consistent with your own notations.
* * *

---

## Author Comment (AC2) · 15 Oct 2017

Dear Professor Roberto Greco,

We are very glad to learn reviewers' recognition on the value of our study and sincerely thank the constructive comments and suggestions to our manuscript (Ref: hess-2017-457). This revision focused on answering the question about the kind of model (recurrent or not) and explaining how the recurrent learning mechanism could be effectively used to predict the multi-step-ahead floods through using inter feedback in the proposed Model 2&3 as the external value (observed flood) is missing. Responses were made to every comment raised by the reviewer, and revisions were incorporated into the revised manuscript. Changes made in the revised manuscript were colored in blue. Modifications in the revised manuscript were made to meet the publication standards required by the HESS.

As known, accurate and robust multi-step-ahead flood forecast during flood season is extremely crucial to reservoir flood control. A modified hybrid learning algorithm, which fuses the Least Square Estimator (LSE) with Genetic Algorithm (GA), is proposed for optimizing the parameters of recurrent ANFIS (R-ANFIS) model to overcome the instability and local minima problems as well as improve model's generalization and robustness. We wish to have an opportunity to share our research methodology and findings with readers through the HESS.

Best Regards,
Fi-John Chang
Professor, Department of Bioenvironmental Systems Engineering
Yanlai Zhou
Post-doctor, Department of Bioenvironmental Systems Engineering
National Taiwan University
Email: changfj@ntu.edu.tw (F. Chang); zyl23bulls@whu.edu.cn (Y. Zhou).
Tel: +886-2-33663452

**Responses to Referee #2:**

*The paper deals with the forecasting of the flood of the TGR using ANFIS model with three versions of this solution. Generally, the paper is not extremely rigorous. For example, the goal is to predict the flood of the dam. But what is the flood of the dam: the flood of the input river (how long upstream to not being influenced by the level in the reservoir)? The flood at the output of the reservoir (to be able to manage flood downstream)?*

**Reply:** We thank you for taking time to read our manuscript and give valuable and constructive comments. As known, the flood forecasting is essential to provide sufficient hydrological information for rational decision making. Accurate and robust multi-step-ahead flood forecast is extremely crucial and desired to TGR flood control and water resources utilization. Based on your very constructive comments and suggestions, we have refined our statements to explain how the recurrent learning mechanism could be effectively used to predict multi-step-ahead floods. We have also modified the Fig 3 to clear represent the recurrent models (Model 2&3) and the traditional (no recurrent) ANFIS model (Model 1). Moreover, we have added (as suggested) the Coefficient of persistence ($G_{bench}$) as one of the evaluation criterion to demonstrate the models' utility of prediction.

We begin with what is the flood of the dam. Indeed, the floods of TGR (Qo) are transformed from the reservoir water level and reservoir outflow, as shown follow.

$$Q_o(t) = \frac{[S(t+1) - S(t)]}{\Delta t} + R(t)$$

where $Q_o(t)$ is the floods of TGR at *t*th time, $R(t)$ is the observed water releases of TGR at *t*th time (from Yichang streamflow gauged station), $S(t)$ and $S(t+1)$ are the reservoir storage (transferred from observed reservoir water level) at *t*th & *t+1*th time, respectively.

Hence, the floods of TGR contain the upstream flows (controlled by streamflow gauged station), indirect runoffs (yielded from rainfalls) and direct rainfalls (directly dropped in the reservoir). All of upstream flow inputs (streamflow gauged stations) used in our case are not being influenced by TGR water level. The observations of XJB reservoir flow, eight flow gauged stations, I & II region rainfalls and TGR flow in the flood season (June 1st to September 30th) from 2003 to 2016 (14 years) with 6 hours' time-step are available for modelling multi-step-ahead TGR inflow forecasts.
* * *
*For the same lack of accuracy, the following sentence has no meaning, mathematically speaking "Vulnerability represents the incompetence of a model to resist the effects of a hostile environment (e.g., the stochastic nature of hydrological variable". It should be better to correct this sentence and to be more accurate and more mathematical in several occasions in the paper (please, see technical comments). Moreover, in my opinion being stochastic is not the real problem. The real problem comes from the ungaussian property of hydrologic signals.*

**Reply:** The constructive comment is sincerely appreciated. We agree and have revised the following statement to be more mathematical description of vulnerability used in this study.

Vulnerability of model refers to the maximum Relative Absolute Error (RAE) of model forecast if RAE is greater than the threshold value of qualified forecast.

$$\text{Vulnerability} = \max_{i=1}^{N}\{V_i\} \tag{9a}$$

$$V_i = \begin{cases} \text{RAE}_i, & if(\quad \text{RAE}_i > \delta\ ) \\ 0 & \text{else} \end{cases} \tag{9b}$$
* * *
*Suffering from the same cause, it is not so easy to know if the models are recurrent or no recurrent. This question is essential because the design of a really recurrent model, where the input is the previously calculated output, is more difficult. It seems to appear that the models are not recurrent (the input seems to be the previous measured discharge plus upstream measured discharges and rains, see table 2). If the model uses previous observed discharge (using Qo(t) to forecast Qf(t+i)) then it is mandatory to evaluate the quality using the persistency criteria in order to appreciate if the model has an added value or not (please see technical comments). Also, all along the paper the variable Q(t) is used. Nevertheless, the variables Qf (forecasted discharge) or Qo (observed discharge) are also defined. Thus what does Q represent?*

**Reply:** We thank you for point-out the indistinct places (models & statements). We clarify the misleading statements and then make all the necessary modification (Table2 & Fig 3). We notice the recurrent learning mechanism is used in our proposed Models (Model 2&3 shown in the revised Fig 3) to predict the multistep ahead floods by using inter feedback (forecasted value).

Sorry for mixed-up of the variable *Q*. We have changed the variable Q(t) into Qo(t) in Table 2. As shown in Table 2, the Gamma test (GT) is used to select model input,

i.e., to determine the best set of inputs from a list of possible inputs for the data-driven models.

Table 2 The optimal lagged variables and input combinations (observed data) used in the three models

| Station | Travel time | Sub-opt | Opt | Ben (I) | Sub-opt | Opt | Ben (II) | Sub-opt | Opt |
|---|---|---|---|---|---|---|---|---|---|
| Xiangjiaba | 48h | Qo(t-4) | Qo(t-4) | Qo(t) | Qo(t) | Qo(t) | Qo(t) | Qo(t) | Qo(t) |
| Hengjiang | 48h | Qo(t-4) | Qo(t-4) | Qo(t) | Qo(t) | Qo(t) | Qo(t) | Qo(t) | Qo(t) |
| Fuxi | 42h | Qo(t-3) | Qo(t-3) | Qo(t) | Qo(t) | Qo(t) | Qo(t) | Qo(t) | / |
| Gaochang | 48h | Qo(t-4) | Qo(t-4) | Qo(t) | Qo(t) | Qo(t) | Qo(t) | Qo(t) | Qo(t) |
| Fushun | 42h | Qo(t-3) | Qo(t-3) | Qo(t) | Qo(t) | Qo(t) | Qo(t) | / | / |
| Chishui | 24h | Qo(t) | Qo(t) | Qo(t) | / | / | Qo(t) | / | / |
| Wucha | 12-18h | Qo(t) | Qo(t) | Qo(t) | / | / | Qo(t) | / | / |
| Beibei | 12-18h | Qo(t) | Qo(t) | Qo(t) | / | / | Qo(t) | / | / |
| Wulong | 6-12h | Qo(t) | Qo(t) | Qo(t) | / | / | Qo(t) | / | / |
| I Rainfall | 42-48h | R(t-1) | R(t-2) | R(t) | / | R(t) | R(t) | / | / |
| II Rainfall | 12-18h | R(t) | R(t) | R(t) | / | / | R(t) | / | / |
| TGR | / | Qo(t) | Qo(t) | Qo(t) | Qo(t) | Qo(t) | Qo(t) | Qo(t) | Qo(t) |
| Horizon | / | t+4 | t+4 | t+8 | t+8 | t+8 | t+12 | t+12 | t+12 |
| *Ratio* | / | 0.0007 | 0.0001 | 0.0115 | 0.0081 | 0.0012 | 0.0121 | 0.0097 | 0.0015 |

We have revised Figure 3 to explain how the recurrent learning mechanism will be used to simulate and predict when external feedback (observed value) is missing, by using inter feedback (simulated value) in Model 2&3.

[Figure]

Figure 3 Framework of three ANFIS modeling approaches (Notes: n denotes the horizon. $\hat{Z}(t)$ and $Z(t)$ denote the simulated and observed inflow of TGR, respectively. $X(t)$ denotes the observed flow or rainfall of upper basin, p and q are the inputs time-lag and feedback time-lag, respectively).
* * *
*The procedure of training is not accurately described (p6 L13: "After implementing an intensive trial-and-error procedure). Let me recall that the paper must be sufficiently accurate to could be reproduced by other people. It is evident that it is not the case.*

**Reply:** The constructive comment is sincerely appreciated. The procedure of training model has been more accurately described and shown in the section 3.2 and Figure 4.
* * *
*P11, what is "the recurrent learning mechanism"?*

**Reply:** Thanks for your constructive comment. The recurrent learning mechanism is used to make multistep predictions in Models 2&3. We have revised Figure 3 to explain how the recurrent learning mechanism is used to predict when external feedback (observed value) is missing and/or unavailable, by using inter feedback (forecasted value) in Model 2&3.
* * *
*The same applies to the procedure of variable selection: the method is not described. But variable selection is essential in data driven models.*

**Reply:** Thanks for your constructive comment. We have used the Gamma test to select model input to determine the best embedding dimension and delay time for time series and to identify the best set of inputs from a list of possible inputs for data-driven model. The Gamma Test only contain the observed inputs and output, not include the simulated or forecasted output of data-driven model. The Gamma test assume that training and testing data are different sample sets in which: (a) the training set inputs are non-sparse in input-space; (b) each output is determined from the inputs by a deterministic process which is the same for both training and test sets; (c) each output is subjected to statistical noise with finite variance whose distribution may be different for different outputs but which is the same in both training and test sets for corresponding outputs. Hence, the Gamma test is separated from data-driven model and can be applied to the procedure of variable selection. The Gamma Test (Koncar 1997) is one of the state-of-the-art input selection techniques and has been successfully used in various hydrological modelling (ex. Chang & Tsai, 2016; Remesan, et al., 2008).

Koncar, N., 1997. Optimisation methodologies for direct inverse neurocontrol, PhD thesis, Department of Computing, Imperial College of Science, Technology and Medicine, University of London.

Chang, F. J., Tsai, M. J., 2016. A nonlinear spatio-temporal lumping of radar rainfall for modeling multi-step-ahead inflow forecasts by data-driven techniques. Journal of Hydrology, 535(2), 256-269.

Remesan, R., Shamim, M. A., Han, D., 2008. Model data selection using gamma test for daily solar radiation estimation. Hydrological processes, 22(21), 4301-4309.
* * *
*Finally, the section presenting results is quite confused and difficult to read. Maybe some Tables and example of predicted signals, at each lead time, should be better to compare the models than the proposed indirect representations. Usually, indirect representations (Fig 6, Fig 7) hide the defect of flood prediction when the peak is not good but the rest of the hydrograph quite well represented. For this reason, it is essential in case of flood prediction to provide an accurate measurement of the quality of the predicted peak, or a representation of the signals.*

**Reply:** We respect the criticism and have tried our best to make all the necessary modification for improving the readability of our manuscript.

We notice that the 6 hours' time-step dataset is used to predict the TGR inflow, thus the horizons t+4, t+8, t+12 are represented for different lead times 1st day, 2nd day and 3rd day. In fact, the residual values (Observation - Forecast) of the three models at horizons t+4, t+8 and t+12 in Fig. 7 not only show the forecasting accuracy of flood peak, but also display the forecasting accuracy of the hydrograph. In Fig.6: the criterion MAE is suitable for measuring the accuracy at medium and low flows, while the criterion RMSE would provide a good measure of the good of fit at high discharge. As suggested, we added the criterion $G_{bench}$, which could estimate the performance of forecasting model by using the observed data shifted backwards by one or more time-lags. The criterion CC could estimate the goodness-of-fit between observed and simulated (forecasted) flood hydrograph. Moreover, we propose a coherent set of evaluation criteria to fully distill the robustness (reliability, vulnerability and resilience) of model based on the criterion RAE.
* * *
*In conclusion, this lack of rigor must be corrected. The question about the kind of model (recurrent or not) must find a response. Only after this response it will be possible to evaluate the quality of the evaluation of the results. Flood forecasting is very difficult and I encourage the authors to deal with more accurately.*

**Reply:** Yes, we agree the recurrent mechanism is one of the most crucial part and have tried our best to make all the necessary modification and clear response to the question about the kind of model (recurrent or not). We also like to notice that we are confident the originality of our proposed methodology in optimizing the parameters of recurrent ANFIS (R-ANFIS) model to overcome the instability and local minima problems as well as improve model's generalization and robustness, which could be very useful and valuable for promoting multi-step-ahead flood forecast as well as for better water

management.
* * *
Specific comments

*- Title: could you justify why the model is qualified of "robust" in the title?*

**Reply:** Thank you for providing this insightful point. We notice that the number of robust knowledge, indicators and assessments studies have been dramatically increased in last few years. In our paper, a coherent set of evaluation criteria is used to fully explore the model's accuracy (MAE, RMSE, CC & $G_{bench}$) and robustness (reliability, vulnerability & resilience). Taking the horizon t+12 (three days ahead) for example, the comparison analysis between R-ANFIS and MR-ANFIS shows that the MR-ANFIS model can further enhance the $G_{bench}$, CC, reliability and resilience by 5.80%, 2.04%, 5.05%, and 3.61%, respectively, as well as decrease the MAE, RMSE, vulnerability by 9.91%, 13.79%, and 9.22%, respectively. Such results evidently promote data-driven model's generalization (accuracy & robustness) and leads to better decisions on real-time reservoir operation during flood season. That is why we used the word "robust" including high accuracy and robustness in the title.
* * *
*- Abstract*

*It is not evident, reading only the abstract to know what are the criteria CC and CE, it is thus necessary to provide, at least, the name of the criteria in the abstract: for example, Ce is the Nash-Sutcliff criterion or the coefficient of determination. And CC is the linear coefficient of correlation.*

**Reply:** Thanks for your friendly suggestion. We have provided the name of the criteria in the abstract as follow.

A coherent set of evaluation criteria is used to fully explore the model's accuracy, i.e., Mean Absolute Error (MAE), Root Mean Square Error (RMSE), Coefficient of persistence ($G_{bench}$) and Coefficient of Correlation (CC) and robustness (reliability, vulnerability & resilience).
* * *
*- Section 3.3. Evaluation criteria.*

*The aim of the paper is to provide prediction. Usually, in this case it is necessary to use a criteria specific to prediction, for example du persistency criteria (Kitanidis, P. K. and Bras, R. L.: Real-time forecasting with a conceptual hydrologic model: 2. Applications and results, Water Resour. Res., 16(6), 1034–1044, doi:10.1029/WR016i006p01034, 1980.). We suggest to authors to calculate also this criteria. This criteria is mandatory when previous measured discharges are used to*

*calculate future discharges, but it is no clear in the paper if previous observed discharges are used or only previous simulated discharges: having exact equations should remove the question.*

**Reply:** Thanks for your constructive comment. We agree and have replaced the criterion CE with Coefficient of persistence ($G_{bench}$) and updated the results of Fig. 6, Fig.10 and Table 3. The criterion $G_{bench}$ is described as,

$$G_{bench} = 1 - \frac{\sum_{i=1}^{N}(Q_f(i)-Q_o(i))^2}{\sum_{i=1}^{N}(Q_o(i)-Q_{bench}(i))^2}, \ G_{bench} \leq 1 \tag{6}$$

where $Q_{bench}(i)$ is the observed data shifted backwards by one or more time-lags. In this case, we use the observed data at time step t as a prediction of the TGR inflow at t+n, and n is the horizon.
* * *
*In table 2 it is not so clearly indicated if the Q(t) of TGR refers to observed or simulated discharge (Qf or Qo)? If it is Qo, then the model is not recurrent at al. The model can simulate a dynamic basin but it is static (finite impulse response).*

**Reply:** Thanks. Sorry for such mistake. We have changed the variable Q(t) into Qo(t) in the text and Table 2.
* * *
*To verify if the model has a utility it is also possible to calculate the Nash criterion of the signal Qo(t+lag). If the Nash criterion of the prediction Qf(t+lag) has a better Nash criterion than the previous one (on Qo), then the predictor is useful; in the contrary case, the model has no interest at all, it is only a model that duplicate, at its output, the received input. This behavior is easy to detect when predicted signals are provided, but it is not the case in this paper. This is a shame.*

**Reply:** The constructive comment is sincerely appreciated. We have replaced the criterion CE with Coefficient of persistence ($G_{bench}$) and updated the results of Fig. 6, Fig.10 and Table 3.
* * *
*Technical corrections*
*P5, L15: correct in Fig 3.*

**Reply:** We thank you for taking time to read our manuscript and give valuable comments for it. Sorry for such mistake. We have changed Fig.2 into Fig.3.
* * *
*Notations in eq 3 are nor fully coherent: i, which is the number of a considered example, appears sometimes in index, sometime in parenthesis.*

**Reply:** Thanks for your friendly suggestion. Sorry for such mistakes. We have unified

the notations of all equations in the paper.
* * *
*P6, L1-2 parameters are not linear or nonlinear. They are used in a linear combination or in a nonlinear function.*

**Reply:** Thanks for your friendly suggestion. We have deleted the descriptions (linear or nonlinear parameters) in the paper.
* * *
*P6, L8: it is necessary to add the equation of the 3 models to express clearly the inputs and outputs variables of the models. Unhopefully, there is a great confusion in the literature about the concept of recurrent (infinite impulse response) and static (Finite impulse response). Could you add the equations?*

**Reply:** The constructive comment is sincerely appreciated. We have revised Figure 3 to explain how the recurrent learning mechanism will be used to simulate and predict when external feedback (observed value) is missing, by using inter feedback (simulated value) in Model 2&3. Please find the revised Figure 3.
* * *
*Eq 9 the criteria RAE is not so good because it could be very high in case of low discharge. It is thus not good when there is very low and very high discharges? In the case of the 3 rivers it is not possible to have our own idea as signals are not provided.*

**Reply:** We agree your constructive comment. As suggested, Eq 9 has been modified. In our paper, the Mean Absolute Error (MAE), Root Mean Square Error (RMSE), Coefficient of persistence ($G_{bench}$) and Coefficient of Correlation (CC) are selected to evaluate the forecasting accuracy of the three models. We also propose a coherent set of evaluation criteria to fully distill the robustness (reliability, vulnerability and resilience) of model. As known, the criterion MAE is suitable for measuring the accuracy at medium and low flows. The criterion RMSE provides a good measure of the good of fit at high discharge. The criterion $G_{bench}$ could estimate the performance of forecasting model by using the observed data shifted backwards by one or more time-lags. The criterion CC could estimate the goodness-of-fit between observed and simulated (forecasted) flood hydrograph. Moreover, we propose a coherent set of evaluation criteria to fully distill the robustness (reliability, vulnerability and resilience) of model.
* * *
*In P9 and others, please used accurate notation: not Q but Qf or Qo. To be consistent with your own notations.*

**Reply:** Thanks. Sorry for such mistake. We have changed the variable Q(t) into Qo(t)

in the text and Table 2.
* * *
The authors would like to thank the editor and anonymous reviewers for their review and valuable comments related to this manuscript.